# Physicians' narratives of communication with patients and their relatives in different phases of the palliative pathway

Bodil J Landstad  ,[1,2] Anett Skorpen Tarberg,[3,4] Marit Kvangarsnes[3,5]

[1]Faculty of Human Science, Mid Sweden University, Ostersund, Sweden
[2]Unit of Research, Education and Development, Östersund Hospital, Ostersund, Sweden
[3]Department of Health Sciences, Norwegian University of Science and Technology, Ålesund, Norway
[4]Møre og Romsdal Hospital Trust, Ålesund, Norway
[5]Unit of Research, Møre og Romsdal Hospital Trust, Alesund, Norway

**Correspondence to**
Professor Bodil J Landstad;
Bodil.Landstad@miun.se

## ABSTRACT

**Objectives** To explore physicians' experiences of the communication with patients and their relatives in the different phases of the palliative pathway.

**Methods** Purposeful sampling was employed to recruit a total of 13 oncologists and general practitioners who engaged in palliative care. A qualitative study with a narrative approach was conducted. Interviews with physicians working in primary and specialist healthcare were conducted via Skype Business in the spring of 2020. The interview guide had open-ended questions with each interview lasting between 35 and 60 min.

**Results** Communication between the physicians, their patients and their relatives was contextual and changed depending on the phase in the palliative pathway. In the first phase, physicians told us that patients and their relatives experienced an emotional shock. Transitioning from the curative to palliative phase was difficult, which emphasised the need for trust through communication. In the middle phase, they revealed that communication about the death process became the priority: what was probably going to happen, the family's role in what was going to happen and perhaps, depending on the illness, any medical decisions that needed to be made. It was important for the physicians to communicate information about the palliative pathway while providing the relatives with knowledge that facilitated any decision making. In the terminal phase, physicians employed a compassionate approach, as bereaved family members needed to process their feelings of guilt and grief.

**Conclusions** The study gives new insight into communication with patients and their relatives during different phases of the palliative pathway, from the physician's perspective. The findings may help physicians improve the quality of communication with patients and their relatives over these vulnerable pathways. The findings also have practical implications in training contexts. The study reveals ethical dilemmas in physicians' communication with patients and their relatives during a palliative pathway.

## INTRODUCTION

Healthcare providers' communication with patients and their relatives in palliative care is crucial.[1] A study from New Zealand found the emotional shock and grief that comes with

## STRENGTHS AND LIMITATIONS OF THIS STUDY

⇒ The narrative approach was appropriate when analysing communication in the different phases of the palliative pathway.
⇒ The interviews gave a rich and thick data related to different phases in the palliative pathway.
⇒ The researchers' different professional positions were a strength.
⇒ The results are transferable to settings with similar health services.

learning of a terminal illness makes it very difficult to hear and process the given health information.[2] Empathy, both as a desirable virtue and as a communication process, has been highlighted in earlier research. Knowing and understanding the patient's perspective can assist clinicians in supporting and guiding the patient through their cancer trajectory.[3]

Communication deficits cause distress for patients and their relatives.[4 5] The majority of formal complaints in healthcare are related to communication.[4 6 7] Palliative care with patients who have limited health literacy poses particular demands on communication.[8] A Swedish study researching experiences in end-of-life communication showed that there existed a communication barrier when the physicians were uncertain about prognoses, which possibly hindered treatment decisions.[9] The complex nature of the transition from curative to palliative treatment can instil uncertainty and diminish the potential for good dialogues.[9]

In this study, we used the definition of palliative care provided by the WHO. Palliative care is an approach that improves the quality of life of patients and their families. Although palliative care is seen as an ethical responsibility, global estimates show that only 14% of patients in need actually receive it.[1] It has been shown to reduce suffering

through the early identification, correct assessment and treatment of pain and other problems, whether physical, psychosocial or spiritual.[1] Palliative care is thus an interdisciplinary approach in which physicians play an important role. As such, this study focused on communication with patients and their relatives by soliciting narratives of communication from the physician's perspective, as this is a highly important component for ensuring quality care.[10] In this context, communication refers to the interpersonal activity of expressing ideas and feelings or providing information; as derived from Latin, communication also refers to 'something you have in common'.[11]

Time and space create the contextual conditions for communication,[12] in which case the physical framework also serves as an important basis. This investigation focused on the type of communication that occurs between physicians, their patients and the patient's relatives in hospitals, home environments or municipal health service institutions. It is an asymmetrical form of communication with a clear purpose. Here, the patient is in a vulnerable position due to both their dependence on the physician's competence and their access to treatment.[12] As a result of this vulnerability, any such communication is linked to the physician's professional role and position within the specific system, making symmetrical communication undesirable and unprofessional. The relative power held by the physician in this role also entails the risk that patients and their relatives will experience dependency and shame.[13] Research has shown that many patients experience shame, guilt and shortcoming in the communication with the physicians in the treatment process.[13]

Research has shown that in acute and critical situations, the need for shared decision making is reduced.[14 15] Trust, therefore, is a central concept in communication between physicians, the patient and their relatives during the palliative pathway. Physicians have both theoretical and practical knowledge whereas patients and relatives usually have neither; therefore, patients are often required to bestow their trust on physicians for decisions on treatment. The patients presuppose that the physician will act in their best interest. This kind of trust-based relationship is, from the physician's side, ultimately necessary to create the needed space to fully use their professional competence.[16]

Sympathy, empathy and compassion are all important for patients at the end stage of life, but research has shown that patients place the highest value on compassion.[17] A systematic review of 31 studies aimed at communication between healthcare professionals and the relatives of patients approaching the end-of-life stage identified seven relevant themes including highlighting deterioration, involvement in decision making, postdecision interactional work, tailoring, honesty and clarity, specific techniques for information delivery and the roles of different healthcare professionals in communication.[18] Patient-centred care and compassion is paramount for patients and relatives during communication in palliative care.[19–22]

Understanding the patient's level of medical knowledge and understanding, that is, their health literacy, as well as any other factors that could impact the patient's ability to understand their situation is critical for the physician to accurately relay information. An example of the consequences of not tailoring information to ensure that patients understanding was shown in a study of cultural knowledge and understanding in the medical field, which found that a lack of cultural understanding among health professionals led to late access or the avoidance of palliative care among indigenous peoples.[2]

The objective of the study is to explore physicians' communication with patients and their relatives in the different phases of the palliative pathways: the early, middle and terminal/bereavement phases. We defined these phases as the initial days following the diagnosis of an incurable disease (the early phase), the time between the early phase and the last weeks or days leading up to death (the middle phase) and the last weeks before death and the time after death (terminal/bereavement phase).[23] We, therefore, posed specific questions in the interview guide to assess the use of communication in the different phases. There is a lack of knowledge about communication in different phases of palliative care from the physician's perspective.

## METHODS

The aim of the study was to explore physicians' experiences of the communication with patients and their relatives in the different phases of the palliative pathway. The study has the following research question: How do physicians experience communication with patients and relatives in different phases of the palliative pathway? We employed a narrative approach and subsequent analysis, which entails different theoretical directions that capture personal and human dimensions of experience over time by considering the relationship between individual experiences and cultural contexts.[24] Here, the narrative approach concerned how physicians perceived communication in the early, middle and terminal/bereavement phases of palliative pathway based on interview data.[25 26] Narrative analysis emphasises the physicians' voice in different phases of the palliative pathway.[26] The study was compiled with the consolidated criteria for reporting qualitative research.[27]

### Study population and recruitment

The physicians were recruited by purposive sampling aimed at capturing the complexity of the palliative pathway.[25 28] The inclusion criteria were physicians with experience from all three phases of care defined in this study.

A contact person recruited a total of 13 physicians working in palliative care, using a direct approach for recruitment through cold calling or face-to-face contact.

| Table 1 | Interview guide |
|---|---|
| | **Questions** |
| 1 | How did you experience being a physician in palliative care? |
| 2 | What is important in communication with patients and relatives in the early phase of the illness? |
| 3 | Can you tell about your communication in the middle/terminal phase of the pathway? |
| 4 | How do you interact with patients and relatives? |
| 5 | What is of importance in communication in the bereavement phase? |
| 6 | Is there anything else you want to share with us? |

To ensure diversity and robustness, we recruited males and females of various age groups, experience levels and medical backgrounds.

## Data collection

We developed an interview guide (table 1) with open-ended questions based on the study's aim and previous research.[29] This facilitated the solicitation of personal experiences related to communication through the progression of palliative pathway.[30] More specifically, the questions were related to how physicians perceived communication during the different phases of the palliative pathway.

Individual narrative interviews were conducted via Skype Business between April and May 2020. Each interview lasted between 35 and 60 min. The interviewer was a researcher as well as cancer nurse, trained to conduct interviews.[27] Saturation was reached on observing that the data gathered through subsequent interviews yielded redundant information.[25] After the 13 interviews, the data were related to the different phases of the palliative pathway.

## Patient and public involvement

None.

## Data analysis

All interviews were audiorecorded and transcribed verbatim. These transcripts were then read to gain an overview of the data.[29] The result section contains stories from 13 physicians that form one narrative. A narrative inquiry research method was chosen as it considers lived experiences and allows for the events that have been of the most significance to be brought forth in the stories participants tell. The interviewed physicians had rich and thick experiences from all phases of the palliative pathway. In narrative research, the researchers are co-creators in shaping the narrative. A narrative is formed by the way we give meaning to our experiences. This means how we understand our own or others' actions as well as the way we organise the events in our minds. How we connect and perceive the consequences of actions over time is of importance for the shaping of the narrative. Meaning making of our experiences can be linked to the past, the present and the future.[29]

During a subsequent content analysis, we searched for a plot based on patterns in the narratives. Specifically, this plot described communication during the early, middle and terminal phases of the palliative pathway.[25 26]

## Ethical consideration

All participants were informed about the study's purpose and procedures, in both oral and written formats. Participation was voluntary and they could withdraw at any time. Further, they were informed that any obtained data would be handled confidentially. The physicians provided informed written consent.

## RESULTS

Oncologists and general practitioners who worked in hospitals or primary care settings provided information about how they supported patients and their relatives during different phases of the palliative pathway (see table 2).

We derived the following three themes related to communication in the (1) early, (2) middle and (3) terminal/bereavement phases, respectively: (1) trusting communication, (2) communication about the death process and (3) communication about processing relatives' guilt and grief. In table 3, we present the analysis process for all three themes. The themes illustrate what the physicians considered as significant in the professional communication with patients and relatives in the different phases of the palliative pathway.

## Trusting communication

The physicians described an asymmetrical relationship in which they were the experts. One said the following:

> Relationship and communication are more important than the patient's choice of treatment.

The narratives revealed that the transition from curative to palliative care was difficult. In addition to the disease-related ailments, patients and their relatives underwent an internalised process of acceptance which often entailed feelings of shock, especially for patients who now realised their limited lifespan. The transition was also often the beginning of a mental process leading to cognition and acceptance of death. Patients had to avail the resources, networks and coping strategies they built up over the course of their lifetime. Time was marked by loss, and patients learnt not to expect improvements.

Physicians expressed they wanted to talk with patients and their relatives together. Here, the goal was to help patients and their relatives develop a common understanding of their situation and the palliative care pathway. One participant said the following:

> We should take the challenge of talking to patients and relatives together because they need to hear what the other person is saying—they need to hear together what we are saying.

**Table 2**  Demographic data on the study participants

| | Participants (n=13) |
|---|---|
| Gender | |
| Men | 10 |
| Women | 3 |
| Age | |
| 41–50 | 7 |
| 51–60 | 4 |
| 61–70 | 2 |
| Place of assignment* | |
| Hospital | 7 |
| Primary care | 7 |
| Private clinic | 1 |
| Education† | |
| General practitioner | 5 |
| Oncologist | 5 |
| Physician | 5 |
| Years of experience | |
| 10–15 | 2 |
| 16–20 | 5 |
| 21–25 | – |
| 26–30 | 4 |
| >30 | 2 |

*Some physicians reported more than one workplace.
†Some physicians worked as both oncologists and general practitioners.

**Table 3**  The analysis process for the three themes

| Quotation | Subtheme | Main theme |
|---|---|---|
| Relationship and communication are more important than the patient's choice of treatment | Paternalistic communication | |
| We should take the challenge of talking to patients and relatives together, because they need to hear what the other person is saying—they need to hear together what we are saying | Joint communication | Trusting communication |
| Any conveyed information must be adapted to their specific needs | Adapted communication | |
| We should take the challenge of talking to patients and relatives together, because they need to hear what the other person is saying—they need to hear together what we are saying. | Common understanding | |
| If we're going to come in and directly inform about prognosis and ask what they want, how they want to die, then I think it's scary—it's too brutal. | Compassion | Communication about the death process |
| We have to accept (situations in which) we do not achieve our treatment goals or a 'good death', which is a value for us. Ain't no cure for death. | Respect for the choices of patients and their relatives | |
| … it would have been beneficial with another day to talk through the situation, or to bring in the family. | To talk through the situation with the family | |
| There is not much that needs to be done, a review of the medical history and what actually happened. | Listening to the families experiencing bereavement | Communication about processing relatives' guilt and grief |
| It was important to provide the relatives with stories they could accept and carry with them when going forward. | To give relatives stories to live with | |

The physicians experienced that patients and relatives did not have enough knowledge to make decisions about medical treatment choices. The physicians emphasised that patients and relatives in such situations needed experienced physicians to guide them in the course of treatment. This was a way of protecting the patients and their relatives from difficult choices.

This presumed responsibility causes stress for the physicians, who were concerned with being responsibility for the recommended choice of treatment. Being experts in this context, physicians are therefore expected to assess the consequences of the available choices to the patient's specific situation and relay that to the patient. As patients and their relatives often experience crisis during this time, the physicians highlighted that 'any conveyed information must be adapted to their [the patient's] specific needs'. Practically, the patients must be informed about the disease, the expected course of the disease and the nature of the treatment during each phase of the pathway. The physicians said it was important to provide this information, by which giving security to patients and their relatives while laying a foundation for appropriate collaboration.

### Communication about the death process
The physicians said that patients and their relatives continually needed information, but that recognition was a gradual process. One said the following:

We have to take the challenge of asking the difficult questions, and we have to choose the right time to do it. I do not think everyone in the early phases has a realisation that allows us to ask how they want to die.

If we're going to come in and directly inform about prognosis and ask what they want, how they want to die, then I think it's scary—it's too brutal.

The physicians conveyed that many patients had ambivalent relationships with death; at times, they would accept their situation, but only a day later express their intent to survive. However, they also said that it was normal and totally acceptable for the patients to have this dual mindset.

Timing is always of essence, especially when communicating with patients, whether to inform or ask questions. Finding the right time to provide patients with information on their disease's development is very important

and that means being cognizant of their patient's mental status and realising when the patient is ready to receive this information. As such, the physicians expressed that they spent substantial amounts of time to get an overview of health literacy among patients and their relatives. In their experiences, there were various prerequisites for adapting to situations and processing the nature of disease development at the individual level.

Patient-centred care was also a recurring theme. Here, the physicians reflected on the importance of avoiding situations in which their professional values, for example, expectations of mastery and a 'good death', prevented them from handling difficult situations and enduring ethical dilemmas. In particular, they expressed the need to respect each family's choices. One physician said the following:

> We have to accept the situations in which we do not achieve our treatment goals or a 'good death', which is a value for us. Ain't no cure for death.

### Communication about processing relatives' guilt and grief

In the terminal/bereavement phase, the physicians said the conflict between efficiency and quality constituted an ethical dilemma in which the demands for efficiency were not always in accordance with the patient's need for time during the process. One example, noted by a physician, was determining how long a patient should stay at the hospital, saying the following:

> … it would have been beneficial with another day to talk through the situation, or to bring in the family.

This recurrent ethical challenge was experienced often by physicians working in hospitals. The terminal phase entails the need for greater involvement of the relatives of patients as it often gives meaning to them during the death process. Having some of the responsibility in caring for their loved ones provided time and space for saying goodbye and experiencing community. There is an importance in listening to the needs of patients and their relatives all the while helping them through the process of acceptance. In this context, it was often beneficial for the physicians to withdraw from the process some, as the patients and their relatives were the true focus. When the process gets to certain point, the focus becomes helping the patients and family come to terms with the situation, as medically there is nothing more that can be done, noted in the following quote:

> There is not much that needs to be done, a review of the medical history and what actually happened.

The process of acceptance for the family is challenging and many relatives often carry guilt moving forward, so 'it was important to provide the relatives with stories they could accept and carry with them when going forward', to help them come to terms with the death. This guilt stems from many areas including that the disease was not detected early enough, or that they had not contributed enough. During bereavement, conversations were thus focused on summarising what occurred over the palliative pathway; the physicians said this helped the relatives end conversations with 'relaxed shoulders'. The atmosphere was often positive, with relatives expressing gratitude and relief. The physicians felt it was important for relatives to talk about any emotional strain they had experienced during the palliative care process. The mourning process was also more conducive in cases where the relatives had contact with physicians early in the process, and thus held continuous dialogues.

## DISCUSSION

The narratives showed various experiences with professional communication during different phases of the palliative pathway. In the early phase, the physicians conveyed that, as the patients and their relatives were in crises mode, trustful communication was crucial. Here, they saw it as important to provide relevant information while working to develop a common understanding. In the middle phase, the need for information was strong with the patients and their families having many questions. Compassion, respect and a common understanding are considered significant in this dialogue and more importantly for the physicians was the timing of the information given. Following this phase into the final and terminal phase, the bereaved relatives needed experiences, interactions and memories that could help them through the acceptance process. As such, conversations during bereavement helped the mourning process and assuaged feelings of guilt and grief.

The transitioning from the curative to palliative phase is difficult for all involved making the dialogue between the physicians and the patient and their families even more important. In palliative care, the goal of the communication is to develop a common understanding of the coming situation that will occur. This dialogue is a way to create trust, which is a central component.[31]

Previous studies have emphasised that patients and their relatives depend on expert knowledge from physicians making the doctor–patient relationship asymmetrical due to professional roles and other factors.[12 16] Research has shown that in acute and critical situations, the need for shared decision making is reduced.[14 15] Physicians must therefore consider issues and needs from the patient's perspective, as they are entrusted with the care of the patient 'in good faith'. In this study, the physicians said their patients conferred trust when allowing them to decide the best course of treatment over the palliative pathway. In other words, patients assume that physicians will act in their best interests. In turn, this creates the space that physicians need to fully use their professional competencies. While this forms a basis for exercising benign power, it is also important to remember that such trust places the patient in a position of vulnerability to negative forms of power.[16] On the other hand, uncertainty about the prognosis may hinder communication

between physicians when attempting to provide information about palliative treatment,[9] which likely imposes a variety of other communicative difficulties. In sum, clear communication is necessary for developing trust during vulnerable health situations.[9 16 17]

We found that timing was important when providing information about disease prognosis. The physicians said they needed to ask difficult questions at the right time, which required an understanding of individual characteristics, including various levels of health literacy among patients and their relatives. The physicians said they did not always achieve their intention of facilitating 'a good death' for the patient, as it was considered important to respect and comply with the wishes of patients and their families. Compassion is widely considered an essential element in palliative care.[20 21 32] A previous study reported that compassion and empathy were preferred over sympathy from the perspective of the patient on palliative care.[17] Compassion, in this case, being the most preferred, as it benefited communication addressing their suffering and needs through relational understanding and action.[17] This definition also emphasises action as an important aspect of compassion, which reflects the practices implemented by this study's physicians, who prioritised the preferences of patients and their relatives over their own professional values during care. Other studies have also highlighted the importance of tailored, clear and patient-centred communication in facilitating death.[18 22] Further, cultural competence is highly important when adapting information to the various levels of health literacy among patients and their relatives.[18] Personal health literacy is defined as the ability to find, understand and use information and services to make health-related decisions and actions for themselves and others.[33] In this study, physicians said it was important to map the health literacy to understand the level of comprehension of the patient and their family members. In doing so, the physician was better adept to change how they spoke to those involved ensuring their message was received and understood as intended.

Following the death of the patient, physicians stated that continued communication during bereavement was important. Relatives needed stories they could relate to within their own lives. In the mourning process, it was helpful when physicians had been involved from early on and were thus able to hold continual dialogues with the relatives. However, research has shown that physicians come into the picture at a later stage in the process and many relatives are not given the opportunity to have these conversations.[32] In sum, both the current and previous findings emphasise that the continuous dialogue during bereavement are important aspects of the palliative care process.

## Conclusion

The study gives new insight into physician's experiences of communication in different phases of the palliative pathway. Trustful communication with patients and their relatives was crucial in the first phase. In the middle phase, communication about the death process is important and, respect for the choices of patients and their relatives were important. A common understanding with the patient and their relatives of the treatment was paramount. In the bereavement phase, talking through the situation with the relatives and giving them stories to live with was of great significance for the physicians.

A conclusion is that the physicians may experience an ethical dilemma in communication with patients and their relatives about the efficiency and quality in the treatment. Conflicts might arise between the physicians and the patient and relatives in, for example, treatment choices. Taking a paternalistic attitude to protect the patient from harm can be morally complicated. In healthcare treatment, the physicians have superior knowledge and insights and thus, in an authoritative position to determine the patient's best interests.[34] A conflict between beneficence and respect for the patient and the relative's autonomy to participate in treatment choices may arise. Physicians should be prepared for such ethical dilemmas. Ethical dilemmas in palliative care should be an important topic of discussion in the community of practices of physicians as well as in medical education.

## Strengths and limitations

This study is limited to the experiences of the physicians. No patient and public involvement is included in the study. The qualitative interview approach provided rich and complex data, as we employed an explicitly theoretical framework, thus enhancing trustworthiness.[12 16 30] Further, direct quotations underpinned the main findings.[27] The researcher's different professional backgrounds, being Registered Nurse and social scientists, aided in understanding the complexity of the obtained narratives. Such intersubjective understanding also improved the study's authenticity.[26] Member checking was done during the interview process by an experienced oncologist. The study was solely conducted in the Norwegian context. Nevertheless, research suggests that our findings may be relevant in other cultural contexts such as New Zealand, Europe and the USA.[2 35]

## Implications of results for practice and further research

In this study, the narrative approach revealed new insights into how physicians communicate with patients and their relatives during different phases of palliative care. In practice, these findings should help improve the overall quality of communication. A close collaboration with relatives in the terminal phase is important for achieving a 'good death'. Additional studies should continue to investigate the issue of cultural dimension of communication in palliative care. It is also of interest to study communication within the team working between health professionals in palliative care. The digital age creates new ways of communicating and it would be interesting to explore the possibilities of this interaction in palliative care.

**Acknowledgements** We express our gratitude to all participating physicians who shared their experiences about communication over the palliative pathway.

**Contributors** BJL, AST and MK contributed to conceptualisation, methodology, validation and investigation. BJL and MK contributed to formal analysis, writing-original draft preparation and writing-review and editing. BJL is responsible for the overall content as the guarantor. All authors have read and agreed to the published version of the manuscript.

**Funding** The authors have not declared a specific grant for this research from any funding agency in the public, commercial or not-for-profit sectors.

**Competing interests** None declared.

**Patient and public involvement** Patients and/or the public were not involved in the design, or conduct, or reporting, or dissemination plans of this research.

**Patient consent for publication** Not applicable.

**Ethics approval** This study involves human participants and was approved. According to the Regional Committee on Medical and Health Research Ethics, this study did not require approval (REK 78067). However, we did receive approval from the Data Protection Officer associated with the Møre og Romsdal Hospital Trust (HMR 2020/397-2) and the Norwegian Centre for Research Data (NSD 131948).

**Provenance and peer review** Not commissioned; externally peer reviewed.

**Data availability statement** No data are available. The interviews are transcribed verbatim in specific Norwegian dialects and it is therefore not appropriate to share the data with other researchers. The dialects are very different from the two Norwegian official languages Bokmål and Nynorsk and therefore not appropriate to share.

**ORCID iD**
Bodil J Landstad http://orcid.org/0000-0001-6558-3129

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
