## [Reviewer comments · BMJ Open]

ARTICLE DETAILS

TITLE (PROVISIONAL)	Physicians' narratives of communication with patients and their relatives in different phases of the palliative pathway
AUTHORS	Landstad, Bodil J.; Kvangarsnes, Marit

VERSION 1 – REVIEW

REVIEWER	Wilson, Margo Memorial University of Newfoundland, Discipline of Emergency Medicine
REVIEW RETURNED	02-Aug-2022

GENERAL COMMENTS	Thank you for the opportunity to review this manuscript. The purpose of this paper was to explore physicians' experiences communicating with patients and their relatives in a palliative care setting. The authors have thoughtfully elucidated the experiences of physicians' communication in a palliative care setting. I thought the concept of communication evolving in a predictable way over time was an especially strong aspect of this paper. Another strength of the paper was a good review of the literature, but I would have liked a discussion of how the existing literature informed the development of the interview guide. The paper could have been improved with some spelling and grammar edits. As well, an explanation of why they chose narrative analysis of the interview transcripts would fortify the paper. I liked the use of quotations in a table format, and this potentially have more impact with the addition of a few more illustrative quotes. In the discussion, the authors recognized of the asymmetry of the patient-physician relationship. However, I wondered about the concept that patients "did not have the prerequisites to make decisions about medical treatment choices". I would like further exploration of this, as I know that in many instances shared decision-making is valued. Is this different in other settings? The concept of forming a narrative for relatives after death was really interesting and I think a key finding of the paper. I think it could be further illustrated with direct quotations. While I liked the general concept, the paper might have been strengthened by a discussion of common issues or barriers experienced while communicating with patients and their circle of care. As well, I was interested in whether or not physician experiences were influenced by years in practice.
---

	The above changes would add some depth to this paper and strengthen the importance of the findings. Overall, this is an interesting look at the physician experiences of communication with patients and relatives in a palliative care setting.
--	--

REVIEWER	Espinoza Suarez, Nataly R Laval University, Laval University
REVIEW RETURNED	22-Nov-2022

GENERAL COMMENTS	Maybe it will be important to make a reference about the clinical pathway/palliative care pathway, or the phases to make clear what we will read later. For example: “ Keywords: Physicians, qualitative research, communication, palliative care, clinical pathway Abstract: Aim: it is not clear which aspect of communication you are looking forward to explore or describe (e.g. quality, type, characteristics, etc.). Methods: Suggested improvement: “A qualitative study with a narrative approach was conducted. We employed purposeful sampling to recruit a total of 13 oncologists and general practitioners who engaged in palliative care, at specialist health care and primary care settings. We used Skype Business software to conduct the interviews in Spring 2020. Here please it will be important to see the analysis done, if a framework was used or the software used for analysis” Results: In this section will be important to know which “phase” or “phases” do the authors refer, or maybe name here the continuum you refer about. Also, it will be better to present the data based on the themes formulated after the analysis. This will be a more organized way to present your results. Conclusion: The way the conclusion is presented now does not show the real contributions of your project. I would suggest focusing more on: what you learn about physicians challenges or facilitators about communication in palliative care, what they feel they need or contribute, in order to reflect the findings of the study. Manuscript: Introduction:  - I would suggest starting the introduction talking about the definition and relevance of palliative care in clinical practice. - On page 5, line 14 to line 24: this part should be reorganized. The first part belongs to the presentation of the study aim. - Page 5 line 28 to 48: is also mixing the study justification and the introduction of the study. This section is a bit confusing because there is a mix of ideas: time and space, settings, patient vulnerability, power and roles. - Page 6 line 5 to 19: there is also a mix of terms and ideas. In this section you talk about emotions and end of life communication. - To reorganize the introduction, I would suggest to:
---

	a) Introduce the definition of palliative care and communication, followed by the importance of communication in patient and clinician relationship b) Later introduce the importance of communication with palliative care. And introduce previous research done about communication in palliative care to present what has been done in the field about this subject. c) Then, present the justification of the study, and the problematic you found that generated the development of this project  - This is a good opportunity to rephrase the aim of the study, maybe as suggested: "To describe physicians' perceptions about communication with patients and their relatives during the different phases of palliative care". - Also, the research question should be moved to the methods section. Methods:  - In the first part of the methods the research question should be introduced as part of the general explanation. - The first subsection should be called: study population and recruitment. In this part is important to mention the inclusion criteria and the number of professionals recruited. Also the sampling method, and the techniques to recruit (e.g. schedule review, phone calls, direct approach, etc.)  - The table 1 should go in the results section. - The second subsection should be called data collection. In this section, should be stated the interview guide. Storytelling is a central feature of narrative research, so this needs to be stated in the methods section. Another relevant data should include the program or how the interviews took place. Also, interviewer data, interview duration. Finally, consider a statement about data saturation. - The patient and public involvement statement can go to maybe study limitations. - For the data analysis section, I would suggest explaining in a clear way what is narrative inquiry research, why it was chosen: "this method takes into account lived experiences and allows for the events that have been of most significance to be brought forth in the stories participants tell." Also, this section needs to reflect if there was member checking, or other methods to ensure trustworthiness. Do not forget to specify the software used for data management. - I would suggest presenting more examples for the table 3 and maybe add this table to the supplements. - Also, in case the team used a theoretical framework to guide the analysis, please describe it at the end of the methods section. And explain why you chose it. - For the table 2: the first question has an orthographic mistake. Results:  - Information from the table 1 should be shortly presented at the beginning of the results section. - About the first theme: The way it is written reflects experiences from patients and their relatives. It also describes what physicians intended to do in these difficult situations and the kind of relationship they have with patients and their relatives.
--	--

	This needs to be rephrased to present the physicians experiences during these hard moments and their experiences during the asymmetrical relationship, maybe it was challenging? Stressful? Hard? From line 32 to 48 on page seven: this paragraph does reflect the physicians' sense of role or responsibility, but again does not reflect their experiences as providers during crises.  - About the second theme: This second theme is also presented from the patient and their relative's scope. From line 28 to line 48 on page 13, this is genuinely related to physicians' experiences about palliative care communication. - About the third theme: The presentation of this theme is closer to a representation to physician experiences, but it can also have components of the description of physician roles. Please make sure to rewrite the theme to answer to the research question. - For the results section it will be beneficial to select themes that reflect physicians' experiences about communication during end of life. Like personal perceptions of feelings, of managing situations, or other instances. - Also, it will be helpful to rename the themes based on the new selection of physicians' experiences. Discussion: The discussion section opening is a good summary of the results, but unfortunately these results do not answer the research question or aim of the study. In fact, the interview guide leads me to think that the information gathered is more related to the description of their activities, the description of their role as physicians and, finally, the description of patients and their relatives needs. The section about trusting communication goes around patients needs and not physician experiences from the results. It is important to work on the rephrasing of the results to make the discussion section accurate. Strengths and limitations: Here you mention an explicit theoretical framework, please this needs to be specified in the methods section. Narrative inquiry is a form of qualitative research and not a theory to guide analysis, so in case there was another framework used please state it in the methods section. The strengths are also, different settings evaluation, different professional background evaluated. The use of researcher from different backgrounds. But in the limitations there is the need to mention that there was no member checking, and other measures to ensure trustworthiness. Quotes need to be reorganized and re-selected in order to answer correctly the research question. In general, the manuscript need a grammar revision before resubmitting the paper.
--	--

REVIEWER	Roodbeen, Ruud
REVIEW RETURNED	10-Dec-2022

GENERAL COMMENTS	Thank you for this opportunity of reviewing the study: 'Physicians' narratives of communication with patients and their relatives in palliative care'. This study explored communication between physicians (oncologists and GPs, n = 13) and their patients/relatives in palliative care. A qualitative study was conducted using a narrative approach. This
---

study addresses an important subject in healthcare, uses a physician's perspective and provides helpful strategies for physicians to improve the quality of communication with patients/relatives. In addition to this, I think there are some major issues that deserve attention before publication is feasible.

Major issues:

- I feel the introduction needs to be revised. In my opinion, the order in which arguments are presented is not always logical and could be improved. Also, the references used in the introduction are all before 2019. More current insights could be added regarding communication between healthcare professionals and patients/relatives in palliative care. In my suggestions on minor revisions, I present some relevant references.
- The authors use the beginning, middle and terminal phases of palliative care. Why these phases? Is there a theoretical substantiation to use these phases? And a corresponding reference? Please clarify.
- I think the discussion section is excellent. The first time I understood some of the methods and results was after reading the first section of the discussion. A lot of insights and some definitions described in that section can be used to clarify other sections in the manuscript. Please revise.
- Please explain and add to the discussion section how healthcare professionals could benefit from the findings from this study. How does all this help them to communicate better? What should they do or change? This could be described more clearly in this section.
- A native speaker should read and revise the entire manuscript, especially the introduction and results section.
- Lastly, please revise the abstract in accordance with all alterations.

Minor issues:

P. 3, line 48: I would use 'healthcare professional' or 'healthcare provider'.

P. 3, line 51: do you mean 'person-centered care'?

P. 3, line 55: in your introduction, you say that: 'In previous research, the Lancet Oncology Commission suggested the need to integrate the tumour-directed and patient-centred approaches, thus focusing on the experiences of individuals suffering from advanced disease and who may require palliative care.²'. So, more focus on experiences of individuals in palliative care, right? Why is that important? And why mentioning it here? Please clarify what you mean with this sentence and/or revise the order of arguments.

P. 4, line 37: you state that patients are vulnerable, and depend on the competence and access to treatment (i.e., power) through the physician. Therefore, you describe symmetrical communication to be undesirable and unprofessional. I'm not sure I agree with that and/or understand what you mean by this section. I think that the roles and division of 'power' between physicians and patients can and should be symmetrical as much as possible, and that during this symmetrical state, desirable communication and care/treatment is feasible, using shared decision-making, without dependence and shame. Or, in other words, symmetrical communication is not undesirable and unprofessional. Please do clarify what you mean with this section.

P. 4, line 55: do you mean 'hinder'?

P. 4, line 57: in your introduction, you say that: 'The complex nature of the transition from curative to palliative treatment may also instill uncertainty and diminish the potential for good dialogues'. I think this statement could use a reference or should be rephrased. I agree that the transition from curative to palliative treatment could instill uncertainty (instill with two ll's), however, this does not automatically mean that good dialogues is diminished.

	P. 5, line 5: I think empathy in communication is important as well for patients, perhaps the work of Liesbeth van Vliet and others might help, for instance:  - van Vliet LM & Back AL. The different faces of empathy in cancer care: From a desired virtue to an evidence-based communication process. Cancer 2021;127:4137-4139 https://doi.org/10.1002/cncr.33833 - van Vliet LM, Epstein AS. The current state of the art and science of patient-clinician communication in progressive disease: Patients' need to know and need to feel known. J Clin Oncol 2014;32:3474-8 https://pubmed.ncbi.nlm.nih.gov/25267758/ P.5, line 30: please provide a definition of 'cultural health literacy'. Also, how about communicating with patients having 'regular' (instead of cultural) limited health literacy in palliative care? Perhaps the studies mentioned here could help: https://journals.plos.org/plosone/article?id=10.1371/journal.pone.0234926 https://bmcpalliatcare.biomedcentral.com/articles/10.1186/s12904-019-0421-x P. 6, line 13: please explain what a palliative pathway is or add a definition/corresponding reference. P. 6, line 20: 77% of participants are male. That is not a mixture. The same applies to age and place of assignment. Please revise. P. 7, line 44: what do you mean with data being 'rich and thick', what does that mean? Please clarify/explain what that is. P. 7, line 56: do you mean 'experience'? P. 8, line 32: what do you mean with a 'holistic impression of the data'? Please explain. P. 8, line 45: 'meaning making' is not proper English, I think. Please revise. P. 12, line 17: doesn't this quote indicate the opposite, that symmetry is desired instead of a description of an asymmetrical relation? Please reconsider/revise. P. 12, line 21: this quote does not match the description of the dual mindset mentioned above the quote. Please reconsider/revise. P. 12, line 34: please explain what 'mapping health literacy' means. P. 12, line 39: a new section should start here. Add spacing before the 'Patient-centered care...' sentence. P. 13, line 17: please revise this sentence, I don't understand it. P. 13, line 34: 'helping them to achieve peace'. Also, please revise the next sentence, I don't understand it. P. 14, line 6: what are 'square shoulders'? P. 16, line 29: which 'explicit theoretical framework' do you refer to? Please explain, and/or add this explanation to the other sections of the manuscript. P. 16, line 43: no other European Western contexts? Please add this and revise.
--	--

VERSION 1 – AUTHOR RESPONSE

Reviewer 1				
1	Another strength of the paper was a good review of the literature, but I would have liked a discussion of how the existing literature informed the development of the interview guide.	We have clarified this at the end of the Introduction.	Introduction	Page 7-8

2	The paper could have been improved with some spelling and grammar edits	The manuscript is language edited by a professional reviewer.	Overall	All pages
3	An explanation of why they chose narrative analysis of the interview transcripts would fortify the paper.	We have added an explanation on why we chose to use narrative analyses.	Methods	Page 8
4	I liked the use of quotations in a table format, and this potentially have more impact with the addition of a few more illustrative quotes.	We have expanded Table 3 with the analysis process for all three themes.	Results	Page 13-15
5	In the discussion, the authors recognized of the asymmetry of the patient-physician relationship. However, I wondered about the concept that patients “did not have the prerequisites to make decisions about medical treatment choices”. I would like further exploration of this, as I know that in many instances shared decision-making is valued. Is this different in other settings?	We have explored the text to explain the difference from other settings.	Discussion	Page 21
6	The concept of forming a narrative for relatives after death was really interesting and I think a key finding of the paper. I think it could be further illustrated with direct quotations.	This has been added in Table 3, as well as in the Result section.	Results	Page 13-15
7	While I liked the general concept, the paper might have been strengthened by a discussion of common issues or barriers experienced while communicating with patients and their circle of care.	We have chosen to include a distinct discussion of our findings. We do not have data to discuss common issues or barriers experienced while communicating with patients and their circle of care.	Discussion	----- ---
8	As well, I was interested in whether or not physician experiences were influenced by years in practice.	We do not have data to explain whether physician experiences were influenced by their number of years in practice.	Discussion	----- ---
Reviewer 2				
1	Maybe it will be important to make a reference about the clinical pathway/palliative care pathway, or the phases to make clear what we will read later.	We have revised the Title. It now reads: Physicians’ narratives of communication with patients and their relatives in different phases of the palliative pathway.	Title	Page 1
2	It is not clear which aspect of communication you are looking forward to explore or describe	We have clarified this in the abstract and in the aim by adding that we are studying	Abstract; Aim	Page 2-3

	(e.g. quality, type, characteristics, etc.).	communication in different phases of the palliative pathway.		
3	Suggested improvement: "A qualitative study with a narrative approach was conducted. We employed purposeful sampling to recruit a total of 13 oncologists and general practitioners who engaged in palliative care, at specialist health care and primary care settings. We used Skype Business software to conduct the interviews in Spring 2020. Here please it will be important to see the analysis done, if a framework was used or the software used for analysis"	We used the framework of communication for the analyses. This is described in the method section (data analysis). We have also expanded Table 3 to show the different steps in the analyses.	Introduction; Method; Results	Page 4-6
4	In this section will be important to know which "phase" or "phases" do the authors refer, or maybe name here the continuum you refer about. Also, it will be better to present the data based on the themes formulated after the analysis. This will be a more organized way to present your results.	The analyses were done according to narrative analyses (the beginning, the middle and the end. ^{29, 26} We have also expanded Table 3 showing the different steps in the analyses. The data underpinning the teams are presented.	Method; Results	Page 8 Page 13-15
5	The way the conclusion is presented now does not show the real contributions of your project. I would suggest focusing more on: what you learn about physicians challenges or facilitators about communication in palliative care, what they feel they need or contribute, in order to reflect the findings of the study.	We have added a Conclusion section where we focus on what we have learnt about physicians' communication in different phases and the ethical dilemmas in communication in the palliative pathway. We have replaced this with the conclusion text in the abstract.	Abstract; Conclusion	Page 2-3 Page 24
6	I would suggest starting the introduction talking about the definition and relevance of palliative care in clinical practice.	We have revised the Introduction section. We have added additional updated references and changed the order of the arguments. We feel that this order in presentation is logical. See Reviewer 3, comment 1.	Introduction	Page 4-8
7	On page 5, line 14 to line 24: this part should be reorganized. The first part belongs to the presentation of the study aim.	We have moved the text and placed it before the aim of the study.	Introduction	Page 7-8
8	Page 5 line 28 to 48: is also mixing the study justification and the introduction of the study. This section is a bit confusing because there is a mix of ideas: time and space, settings, patient vulnerability, power and roles.	We have revised the Introduction section.	Introduction	Page 4-8
9	Page 6 line 5 to 19: there is also a mix of terms and ideas. In this section you talk about emotions and end of life communication. -	We have revised the Introduction section. We have added additional updated references and changed the order of the arguments. We feel the current	Introduction	Page 4-8

	To reorganize the introduction, I would suggest to: a) Introduce the definition of palliative care and communication, followed by the importance of communication in patient and clinician relationship b) Later introduce the importance of communication with palliative care. And introduce previous research done about communication in palliative care to present what has being done in the field about this subject. c) Then, present the justification of the study, and the problematic you found that generated the development of this project	order of presentation is logical and in accordance with the suggestions.		
10	This is a good opportunity to rephrase the aim of the study, maybe as suggested: "To describe physicians' perceptions about communication with patients and their relatives during the different phases of palliative care".	We have rephrased the aim according to the suggestions from all reviewers.	Aim; Methods	Page 2 Page 8
11	Also, the research question should be moved to the methods section.	We have moved the research question to the method section.	Methods	Page 8
12	In the first part of the methods the research question should be introduce as part of the general explanation.	This is done.	Methods	Page 8
13	The first subsection should be called: study population and recruitment. In this part is important to mention the inclusion criteria and the number of professional recruited. Also the sampling method, and the techniques to recruit (e.g. schedule review, phone calls, direct approach, etc.)	We have added text to clarify the recruitment process.	Methods	Page 8-9
14	The table 1 should go in the results section.	Done	Results	Page 13-15
15	The second subsection should be called data collection. In this section, should be stated the interview guide. Storytelling is a central feature of narrative research, so this needs to be stated in the methods section. Another relevant data should include the program or how the interviews took place. Also, interviewer data, interview duration. Finally, consider a statement about data saturation.	We have changed the order of the text. We feel the current order of presentation is logical and in accordance with the suggestions.	Methods	Page 10

1 6	The patient and public involvement statement can go to maybe study limitations.	Done	Methods	Page 24
1 7	For the data analysis section, I would suggest explaining in a clear way what is narrative inquiry research, why it was chosen: "this method takes into account lived experiences and allows for the events that have been of most significance to be brought forth in the stories participants tell." Also, this section needs to reflect if there was member checking, or other methods to ensure trustworthiness.	Member checking was done by an experienced oncologist during the interview process. We have added that in the Strength and limitations section.	Strengths and limitations	Page 25
1 8	I would suggest presenting more examples for the table 3 and maybe add this table to the supplements.	Done	Results	Page 13-15
1 9	Also, in case the team used a theoretical framework to guide the analysis, please describe it at the end of the methods section. And explain why you chose it.	See reviewer 2, comment 3.	Methods	Page 4-6
2 0	For the table 2: the first question has an orthographic mistake.	Corrected	Results	Page 10
2 1	Information from the table 1 should be shortly presented at the beginning of the results section.	Done	Results	Page 13
2 2	About the first theme: The way is written reflects experiences from patients and their relatives. It also describes what physicians intended to do in these difficult situations and the kind of relationship they have with patients and their relatives. This needs to be rephrased to present the physicians experiences during these hard moments and their experiences during the asymmetrical relationship, maybe it was challenging? Stressful? Hard? From line 32 to 48 on page seven: this paragraph does reflect the physicians' sense of role or responsibility, but again does not reflect their experiences as providers during crises.	We have clarified the content and the analyses of each theme. See Table 3. The themes illustrate what the physicians saw as significant in their communication with patients and relatives in the different phases of the palliative pathway. The interview guide would have required another set of questions if we wanted to include the physicians experiences as providers during crises in a phenomenological sense.	Results	Page 13-15
2 3	About the second theme: This second theme is also presented from the patient and their relative's scope. From line 28 to line 48 on page 13, this is genuinely related to physicians' experiences about palliative care communication.	See above comment.	Results	Page 13-15

2 4	About the third theme: The presentation of this theme is closer to a representation to physician experiences, but it can also have components of the description of physician roles. Please make sure to rewrite the theme to answer to the research question.	See above comment.	Results	Page 13-15
2 5	For the results section it will be beneficial to select themes that reflect physicians' experiences about communication during end of life. Like personal perceptions of feelings, of managing situations, or other instances.	We have clarified the content and the analyses of each theme. See Table 3.	Results	Page 13-15
2 6	Also, it will be helpful to rename the themes based on the new selection of physicians' experiences. Quotes need to be reorganized and re-selected in order to answer correctly the research question.	We have clarified the content and the analyses of each theme. See Table 3.	Results	Page 13-15
2 7	The discussion section opening is a good summary of the results, but unfortunately these results do not answer the research question or aim of the study. In fact, the interview guide leads me to think that the information gathered is more related to the description of their activities, the description of their role as physicians and, finally, the description of patients and their relatives needs.	We have revised the aim and the research question. We believe that all our adjustments in the entire text make the article more consistent.	Discussion	Page 20-23
2 8	The section about trusting communication goes around patients needs and not physician experiences from the results. It is important to work on the rephrasing of the results to make the discussion section accurate.	See comment 22.	Discussion	Page 13-15
2 9	Here you mention an explicit theoretical framework, please this needs to be specified in the methods section. Narrative inquiry is a form of qualitative research and not a theory to guide analysis, so in case there was another framework used please state it in the methods section.	We have revised the text. It now reads: The qualitative interview approach provided rich and complex data, as we employed a theoretical framework of communication and trust to reveal theoretical perspectives for analysis, thus enhancing trustworthiness. ^{12 16,31}	Strengths and limitations	Page 24
3 0	The strengths are also, different settings evaluation, different professional background evaluated. The use of researcher from different backgrounds. But in the limitations there is the need to mention that there was no	Member checking was done by an experienced oncologist during the interview process. We have added the in the Strength and limitation section.	Strengths and limitations	Page 25

	member checking, and other measures to ensure trustworthiness.			
3 1	In general, the manuscript need a grammar revision before resubmitting the paper.	The manuscript language has been edited by a professional reviewer.	Overall	All pages
Reviewer 3				
1	I feel the introduction needs to be revised. In my opinion, the order in which arguments are presented is not always logical and could be improved. Also, the references used in the introduction are all before 2019. More current insights could be added regarding communication between healthcare professionals and patients/relatives in palliative care. In my suggestions on minor revisions, I present some relevant references.	We have revised the text with updated references as well as changed the order of the arguments. We feel that the current order of presentation is logical. See reviewer 2, comment 6.	Introduction	Page 4-8
2	The authors use the beginning, middle and terminal phases of palliative care. Why these phases? Is there a theoretical substantiation to use these phases? And a corresponding reference? Please clarify.	This is clarified methodologically and theoretically in the text.	Introduction; Methods	Page 7 Page 8
3	I think the discussion section is excellent. The first time I understood some of the methods and results was after reading the first section of the discussion. A lot of insights and some definitions described in that section can be used to clarify other sections in the manuscript. Please revise.	We have used some of the definitions described in the Discussion to clarify other sections in the manuscript.	Introduction; Methods	Page 4-8 Page 8
4	Please explain and add to the discussion section how healthcare professionals could benefit from the findings from this study. How does all this help them to communicate better? What should they do or change? This could be described more clearly in this section.	We have explored this in the section Conclusion.	Conclusion	Page 24
5	A native speaker should read and revise the entire manuscript, especially the introduction and results section.	The manuscript language has been edited by a professional reviewer.	All over	Over all
6	Lastly, please revise the abstract in accordance with all alterations.	Done	Abstract	Page 2-3
	Minor issues			

9	P. 3, line 48: I would use 'healthcare professional' or 'healthcare provider'.	We have changed to health care providers.	Introduction	Page 4
10	P. 3, line 51: do you mean 'person-centered care'?	Yes – this is a misspelling. This is now corrected.	Introduction	Page 4
11	P. 3, line 55: in your introduction, you say that: 'In previous research, the Lancet Oncology Commission suggested the need to integrate the tumour-directed and patient-centred approaches, thus focusing on the experiences of individuals suffering from advanced disease and who may require palliative care.2'. So, more focus on experiences of individuals in palliative care, right? Why is that important? And why mentioning it here? Please clarify what you mean with this sentence and/or revise the order of arguments.	We have removed the sentence.	Introduction	-----
12	P. 4, line 37: you state that patients are vulnerable, and depend on the competence and access to treatment (i.e., power) through the physician. Therefore, you describe symmetrical communication to be undesirable and unprofessional. I'm not sure I agree with that and/or understand what you mean by this section. I think that the roles and division of 'power' between physicians and patients can and should be symmetrical as much as possible, and that during this symmetrical state, desirable communication and care/treatment is feasible, using shared decision-making, without dependence and shame. Or, in other words, symmetrical communication is not undesirable and unprofessional. Please do clarify what you mean with this section.	We have added text to explain what we mean-.	Introduction	Page 4
13	P. 4, line 55: do you mean 'hinder'?	This is now corrected.	Introduction	Page 4
14	P. 4, line 57: in your introduction, you say that: 'The complex nature of the transition from curative to palliative treatment may also instill uncertainty and diminish the potential for good dialogues'. I think this statement could use a reference or should be rephrased. I agree that the transition from	We have re-phrased the sentence and added a reference. "Instill" is now corrected.	Introduction	Page 5

	curative to palliative treatment could instill uncertainty (instill with two ll's), however, this does not automatically mean that good dialogues is diminished.			
1 5	P. 5, line 5: I think empathy in communication is important as well for patients, perhaps the work of Liesbeth van Vliet and others might help, for instance: - van Vliet LM & Back AL. The different faces of empathy in cancer care: From a desired virtue to an evidence-based communication process. Cancer 2021;127:4137-4139 https://doi.org/10.1002/cncr.33833 - van Vliet LM, Epstein AS. The current state of the art and science of patient-clinician communication in progressive disease: Patients' need to know and need to feel known. J Clin Oncol 2014;32:3474-8 https://pubmed.ncbi.nlm.nih.gov/25267758/	We have added van Vliet's both references in the text and written a short text about the different faces of empathy in cancer care.	Introduction	Page 7
1 6	P.5, line 30: please provide a definition of 'cultural health literacy'. Also, how about communicating with patients having 'regular' (instead of cultural) limited health literacy in palliative care? Perhaps the studies mentioned here could help: https://journals.plos.org/plosone/article?id=10.1371/journal.pone.0234926 https://bmcpalliatcare.biomedcentral.com/articles/10.1186/s12904-019-0421-x	We have replaced the wording Cultural health literacy with Cultural understanding. We have also added text about the importance of communicating with patients having limited health literacy in palliative care.	Introduction	Page 7
1 7	P. 6, line 13: please explain what a palliative pathway is or add a definition/corresponding reference.	We have added a definition of what a palliative pathway is.	Methods	Page 7
1 8	P. 6, line 20: 77% of participants are male. That is not a mixture. The same applies to age and place of assignment. Please revise.	We have deleted the wording "a mixture of".	Methods	Page 9
1 9	P. 7, line 44: what do you mean with data being 'rich and thick', what does that mean? Please clarify/explain what that is.	We have added following text: Saturation was reached upon observing that the data gathered through subsequent interviews yielded redundant information	Methods	Page 11
2 0	P. 7, line 56: do you mean 'experience'?	Corrected	Methods	Page 13

2 1	P. 8, line 32: what do you mean with a 'holistic impression of the data'? Please explain.	We have replaced the wording "holistic impression" with "an overview" of the data.	Methods	Page 11
2 2	P. 8, line 45: 'meaning making' is not proper English, I think. Please revise.	It is commonly used in qualitative method. It is about interpretation of situations in the light of previous knowledge and experiences.	Methods	Page 11
2 3	P. 12, line 17: doesn't this quote indicate the opposite, that symmetry is desired instead of a description of an asymmetrical relation? Please reconsider/revise.	We have changed the order of the text to strengthen the content of this citation.	Results	Page 18
2 4	P. 12, line 21: this quote does not match the description of the dual mindset mentioned above the quote. Please reconsider/revise.	We have changed the order of the text.	Results	Page 18
2 5	P. 12, line 34: please explain what 'mapping health literacy' means.	We have replaced "mapping" with "to get an overview of".	Results	Page 18
2 6	P. 12, line 39: a new section should start here. Add spacing before the 'Patient-centered care...' sentence.	Done	Results	Page 18
2 7	P. 13, line 17: please revise this sentence, I don't understand it.	We have removed some text in the citation to make the meaning clearer.	Results	Page 19
2 8	P. 13, line 34: 'helping them to achieve peace'. Also, please revise the next sentence, I don't understand it.	We have replaced "achieve peace" with "through the process of acceptance".	Results	Page 19
2 9	P. 14, line 6: what are 'square shoulders'?	We have changed to "relaxed shoulders".	Discussion	Page 20
3 0	P. 16, line 29: which 'explicit theoretical framework' do you refer to? Please explain, and/or add this explanation to the other sections of the manuscript.	See reviewer 2, comment 29.	Discussion	Page 24
3 1	P. 16, line 43: no other European Western contexts? Please add this and revise.	We have replaced England with Europe.	Discussion	Page 25

VERSION 2 – REVIEW

REVIEWER	Wilson, Margo Memorial University of Newfoundland, Discipline of Emergency Medicine
REVIEW RETURNED	28-Apr-2023

GENERAL COMMENTS	Thank you for the opportunity to review this study. It is an interesting view on palliative care physicians' communications.
--

REVIEWER	Espinoza Suarez, Nataly R Laval University, Laval University
-----------------	---

REVIEW RETURNED	17-Apr-2023
GENERAL COMMENTS	Congratulations, this is very relevant research. Thanks for addressing the previous review commentaries and for your kind responses.
REVIEWER	Roodbeen, Ruud
REVIEW RETURNED	19-Apr-2023
GENERAL COMMENTS	None.

VERSION 2 – AUTHOR RESPONSE

Reviewer			
1	The paper would be improved by either eliminating or further exploration of the “ethical dilemma”. I did not feel there was enough provided in the results to justify the conclusions in this area.	We have added text to explore the ethical dilemmas in the conclusion part and added one more reference. Ethical dilemmas are described in theme 1 (Trusting communication: Paternalistic communication) and (Communication about the death process: Respect for the choices of patients and their relatives, and the conclusion is underpinned by data from the interviewees.	Conclusion Page 19
2	One other minor note regarding the participants – how do GPs differ from physicians? Also is there any mention of rural versus urban sampling? Are all participants from one geographic location	We have recruited physicians from all three phases of care defined in the study. We have not studied experiences between GP’s practicing in rural vs. urban areas. We have added following text in the methodological section: The recruited physicians were from both urban and rural areas in the middle part of Norway.	Methods Page 6
3	There are minor copyediting errors.	The manuscript is language edited by a professional reviewer. Can you please point out the copyediting errors?	? ?

4	One other area to address is why to include only physicians' perspectives. Is there something to be added to the literature? At times it may be difficult for providers to self-identify problems with their own communication. I feel this should be further addressed in the limitations.	This is already addressed in the limitation of the study: "This study is limited to the experiences of the physicians. No patient and public involvement is included in the study." We have added following text: Physicians communication with patients and relatives should also be studied from the perspectives of patients and relatives as well as nurses. It may be difficult for physicians to self-identify problems with their own communication. It will give a more comprehensive understanding if the topic is illuminated from different perspectives.	Strengths and Limitations	Page 19
---	---	--	---------------------------	---------

VERSION 3 – REVIEW

REVIEWER	Wilson, Margo Memorial University of Newfoundland, Discipline of Emergency Medicine
REVIEW RETURNED	24-May-2023
GENERAL COMMENTS	Thank you for addressing all the comments and many congratulations.

VERSION 3 – AUTHOR RESPONSE